# Survivin as a Therapeutic Target for the Treatment of Human Cancer

**DOI:** 10.3390/cancers16091705

**Published:** 2024-04-27

**Authors:** Qiang Wang, Mark I. Greene

**Affiliations:** 1Department of Medicine, Cedars-Sinai Medical Center, Los Angeles, CA 90048, USA; 2Department of Pathology and Laboratory Medicine, Perelman School of Medicine, University of Pennsylvania, Philadelphia, PA 19104, USA

**Keywords:** survivin, cancer, mitosis, apoptosis

## Abstract

**Simple Summary:**

Survivin is overexpressed in a wide variety of human cancers and is associated with increased chemotherapy resistance, recurrence, and shorter patient survival. Although survivin was first identified as an inhibitor of apoptosis based on its sequence homology, recent studies show that survivin primarily plays a role in the regulation of cell division, as a component of the chromosome passenger complex, which can be localized to the centromere, the spindle midzone, or midbody in a cell cycle-dependent manner. Disruption of survivin function generally leads to mitotic catastrophe and is associated with elevated levels of aneuploidy. In contrast, ectopic expression of survivin can promote cell survival under certain conditions and is implicated in mediating resistance to various cancer drugs, possibly by interactions with molecules that modulate apoptotic pathways. A potential role of survivin in the regulation of mitochondrial functions and processes of autophagy has emerged. Because of its prevalent overexpression in cancer and very limited expression in normal tissues, survivin has been proposed as an ideal therapeutic target, and various approaches have been investigated for survivin inhibition. Here we provide a critical review of our current understanding of the role of survivin in promoting malignancy and strategies for the development of survivin-targeted therapy for cancer.

**Abstract:**

Survivin was initially identified as a member of the inhibitor apoptosis (IAP) protein family and has been shown to play a critical role in the regulation of apoptosis. More recent studies showed that survivin is a component of the chromosome passenger complex and acts as an essential mediator of mitotic progression. Other potential functions of survivin, such as mitochondrial function and autophagy, have also been proposed. Survivin has emerged as an attractive target for cancer therapy because its overexpression has been found in most human cancers and is frequently associated with chemotherapy resistance, recurrence, and poor survival rates in cancer patients. In this review, we discuss our current understanding of how survivin mediates various aspects of malignant transformation and drug resistance, as well as the efforts that have been made to develop therapeutics targeting survivin for the treatment of cancer.

## 1. Survivin and Cancer

Survivin overexpression has been found in most human cancers and is associated with poor prognosis [1,2,3,4,5,6,7,8]. In particular, high levels of survivin expression are linked with the metastasis of various forms of human cancers, including the presence of circulating tumor cells [9,10,11,12,13]. Overexpression of survivin can facilitate the bypassing of cell cycle checkpoints and promote the survival of aneuploid cells [14,15]. Survivin renders cancer cells resistant to radiation [16,17].

While survivin plays an essential role in early embryogenesis [18,19], its expression levels are very low or undetectable in adult tissues and are usually restricted to stem cells and progenitor cells [15,20,21,22,23]. As shown in conditional knockout mice, survivin is required for T-cell development and homeostasis and triggers p53-dependent cell cycle arrest [24,25]. Similarly, survivin also plays an essential role in B-cell expansion [26]. Moreover, survivin is essential for pancreatic beta cell expansion [27,28,29], early brain development [30], and intestinal epithelial progenitor cells [31].

Survivin, encoded by the *BIRC5* gene, is a polypeptide of 142 amino acid residues (Figure 1). The transcription of the *BIRC5* gene is mediated through a TATA-less promoter that contains multiple Sp1 sites, a CpG island subjected to potential epigenetic modifications, and cell cycle-dependent element (CDE)/cell cycle homology region (CHR) boxes that mediate cell cycle-dependent gene expression [32,33]. Survivin expression can be upregulated by multiple pathways that are commonly activated in human cancers, such as EGFR, p185Her2/neu, PI-3 kinase, MAPK, NF-κB, and mTOR [34,35,36,37,38]. The transcriptional events from the survivin promoter can also be modulated by Wnt/β-catenin [39], notch [40], YAP [41], and hedgehog signaling pathways [42,43]. In addition, survivin is regulated by Forkhead box m1 (Foxm1), a transcriptional factor critical for G1/S transition and mitotic progression [44]. Conversely, survivin expression can be downregulated by several tumor suppressors, such as TP53, PTEN, Rb, and BRCA1 [45,46,47,48], which are frequently silenced in human cancers.

At least five splicing variants of survivin have been described, which include survivin-ΔEx3 [49,50], survivin-2α [51], survivin-3α [52], survivin-2B [53], and survivin-3B [54]. The differential splicing events lead to the generation of proteins with a shortened BIR domain or truncated polypeptides that do not have the intact NES or the coiled-coil region in the c-terminus, which can exhibit distinct localization patterns. For example, survivin-Ex3 lacks the NES but contains a distinct bipartite nuclear localization signal (NLS) that mediates its localization to the nucleus [49]. In contrast, in survivin-2B, the BIR motif is interrupted by an in-frame insertion of a cryptic exon, generating a protein predominantly localized to the cytoplasm. Survivin-Ex3 and survivin-2B showed reduced affinity to CPC and cannot compensate for the loss of survivin functions [54]. The survivin splicing variants have been reported to be associated with certain transformed phenotypes and clinical outcomes [55]. However, it should be noted that these studies were all focused on the detection of the RNA forms of the splicing variants [49,50,52,53,54]. There has been limited success in the development of antibodies specific for the survivin variants [51,53,54]. Overall, the protein forms of the variants at the endogenous level remain to be convincingly demonstrated in cell lines or pathological specimens. Moreover, it is controversial whether the splicing variants play an important role in mediating cellular functions [54,56,57].

## 2. Role of Survivin in Cell Division

Survivin plays an essential in cell division, and a loss of survivin leads to mitotic failure and cell death [32,58,59] (Figure 2A). Survivin participates in mitotic checkpoint regulation as a component of the chromosomal passenger complex (CPC) that also includes aurora kinase B, INCENP, TD-60, and borealin [60]. The BIR domain of survivin can interact with a phosphorylated form of histone H3 (Histone H3T3ph), which is mediated by the histone H3-associated kinase HASPIN and required for recruitment of the other CPC proteins to the inner centromere [61,62,63]. Interference with the survivin–histone H3T3ph interaction leads to mislocalized aurora kinase B and mitotic defects [61,62,63]. Upon entry into mitosis, survivin is localized to the centromere region in a manner that is dependent on the inner centromere protein (INCENP) and cooperates with Aurora kinase B and other CPC proteins to modulate spindle formation and proper chromosome alignment [64,65]. Thus, the essential role of survivin in CPC assembly and mitotic progression, mediated by the BIR domain, may be exploited for the development of survivin-targeting therapeutics.

When chromosome segregation occurs at the initiation of anaphase, survivin is separated from the centromere but remains in the spindle midzone, subsequently becoming associated with the midbody. The molecular mechanism involved in the relocation of survivin to the midbody is not well understood. Nonetheless, survivin has been shown to interact with non-muscle myosin II, and midbody-localized survivin is implicated in playing a role in the formation of the contractile ring during cytokinesis [66]. Survivin abnormality is commonly accompanied by aneuploidy [32,58,59,67,68], which supports the notion that it plays an essential role in the regulation of mitotic checkpoints and cytokinesis. Notably, a loss of p53 function is required for re-entry into the cell cycle following the depletion of survivin [69,70].

## 3. Role of Survivin in Apoptosis

Survivin was first identified as a member of the Inhibitor-of-Apoptosis (IAP) protein family, also known as the Baculoviral IAP repeat-containing (BIRC) proteins, based on the presence of a Baculovirus IAP Repeat (BIR) in the N-terminus [32,58] (Figure 1). The IAP family proteins share the common feature of having at least one BIR domain, which consists of ~70 amino acid residues and is involved in mediating protein–protein interactions [71]. While ablation of survivin can lead to apoptosis, overexpression of survivin can protect cells from apoptosis under various experimental conditions [32,58,72,73] (Figure 2B).

Some IAP family proteins can inhibit apoptosis by directly binding to the activated form of caspase and blocking its enzymatic activities [74,75]. For example, the second BIR domain and a linker region of XIAP can directly bind to caspase and hinder substrate access [76,77,78,79]. In addition, some IAPs also contain either a RING domain—which functions as an E3 ubiquitin ligase—or a ubiquitin-associated domain, which mediates the ubiquitin-mediated proteolytic degradation of caspase [80]. In comparison, survivin contains only a BIR domain, and, to date, no compelling evidence is available to show that survivin can directly bind and inhibit caspase activities (Figure 2B).

Survivin has been reported to bind to the mitochondrial protein DIABLO/SMAC [81,82]. DIABLO/SMAC can potentiate certain forms of apoptosis by blocking the action of IAPs and thereby activating caspases [83,84]. Thus, it has been proposed that survivin may inhibit apoptosis by neutralizing the capacities of DIABLO/SMAC to promote apoptotic signaling (Figure 2B). However, it remains to demonstrate that, as endogenous expression levels, the survivin-DIABLO interactions indeed participate in protection from apoptosis [85].

## 4. Role of Survivin in Mitochondrial Function and Autophagy

Survivin has been shown to inhibit mitochondrial-dependent apoptotic events [73]. It has been reported that the N-terminus of survivin contains a mitochondria-targeting sequence that can direct protein localization to the mitochondria when fused with a reporter protein [86] (Figure 1). Interestingly, survivin can be detected in the mitochondrial fraction in cancer cells but not in non-transformed cells [87,88], which suggests that the role of survivin in mitochondria function may be dependent on cell content, including the genetic or epigenetic background. The overexpression of mitochondria-targeted survivin can protect cells from apoptosis and enhance transformation (Figure 2C), which may involve its binding to another IAP family member XIAP [87,89]. The localization of survivin to the mitochondria can also promote cancer cell invasion and metastasis [90]. The overexpression of survivin appears to alter the dynamic of mitochondrial fission and fusion [91] or inhibit mitophagy [88].

Paradoxically, it has been reported that both the knockdown [90] and overexpression of survivin [88,91] can disable mitochondrial functions and reduce oxidative phosphorylation in cancer cells. It should be noted that the studies on mitochondrial survivin were conducted using a fusion protein of survivin with the mitochondrial targeting sequence of cytochrome c, and the knockdown approach used in the study does not specifically target the mitochondrial pool of the protein [87,88,90]. Clearly, additional work is needed to unravel the mechanism and the biological significance of mitochondrial localization of survivin.

Survivin has been reported to physically interact with several proteins that are involved in autophagy (Figure 2D), a process by which cancer cells can adapt to physiological or pathological challenges by degrading and recycling subcellular components of the cell [92]. For example, beclin, a key regulator of autophagy [93], was found to bind to survivin, which may be involved in regulating survivin protein levels [94]. Intriguingly, ATG5, a protein known for its role in the formation and elongation of autophagosome [95], was reported to form a complex with survivin in the nucleus upon exposure to DNA damage, leading to mitotic catastrophe in an autophagy-independent manner [96]. Conversely, the interaction between survivin and ATG5/ATG12 may also impact autophagy-mediated events responsive to DNA damage [97]. The induced ectopic expression of survivin appears to be required for autophagy induced by the inhibition of glycolysis [98]. However, these observations were made only in cell lines with forced overexpression of survivin. It remains to determine whether survivin directly participates in the regulation of autophagic processes.

## 5. Survivin Localization

Survivin contains a nuclear export signal (NES) that binds to the nuclear export receptor Crm1, which is required for survivin cytoplasmic localization during the interphase [49,99,100,101]. Alteration of the NES, which is located between BIR and the c-terminal coiled-coil structure, can disrupt the nuclear export and localization of survivin to the centromere or to the midbody but not its homodimerization or binding to several CPC proteins [100,101,102]. The NES mutations caused a shift from a cytoplasmic localization pattern to a nuclear one, which is associated with the loss of survivin function to protect cells from apoptosis induced by genotoxic damage or external stimulus [99,101]. In addition, the nucleus-directed survivin protein appears to enhance cancer cell sensitivity to apoptosis [103,104].

These observations led to the notion that cytoplasmic survivin is primarily involved in protection from apoptosis. However, it should be noted that the survivin localization patterns in the cytoplasm or nucleus may not be a reliable biomarker for clinical outcomes, as it has been linked to both favorable [8,105,106,107] and unfavorable prognosis of cancer patients [108,109,110,111,112,113].

## 6. Survivin Protein Structure and Post-Translational Modification

The survivin protein structure in the form of a homodimer has been determined by both crystallography [114,115,116] and nuclear magnetic resonance (NMR) [117]. The N-terminal BIR domain consists of a three-stranded β-sheet and four α-helices, with a zinc-binding fold, and the survivin protein forms a bow-tie-shaped dimer via part of the N-terminal region and the linker region between the BIR domain and the C-terminal helix [114,115]. Notably, the ubiquitination of survivin on several lysine residues within the BIR domain is implicated in playing a role in modulating its localization to the centromere [118].

The structure of survivin that features a heteromeric complex formed with borealin and INCENP has also been resolved [119]. In this structure, the C-terminus of survivin, which contains an extended α-helical coiled-coil domain, forms a three-helical bundle with elements of borealin and INCENP in 1:1:1 stoichiometry [119]. These interactions are essential for the central spindle and midbody localization of the complex. More recently, the crystal structure of survivin with the N-terminal tail of histone H3 has also been reported, which identified structural features in the BIR domain that are important for binding to histone H3T3ph [120].

An accumulating body of evidence shows that survivin can be regulated by phosphorylation. Phosphorylation of survivin at threonine 34 by CDK1 has been shown to be important for its anti-apoptotic role [121,122]. In addition, phosphorylation by PKA at serine 20 is also involved in protection against apoptosis by mediating the interaction with XIAP [89]. Moreover, survivin can be phosphorylated by aurora B at threonine 117 and negatively regulates its localization to the centromere region and function in mitosis [123]. CK2 can phosphorylate survivin at threonine 48 in the BIR domain, which is critical for its mitotic and antiapoptotic functions [124]. Furthermore, PLK1 phosphorylates survivin at threonine 20, which seems to be involved in chromosome orientation during mitosis [125].

Survivin is subject to other forms of post-translational modification. For example, survivin can be acetylated at lysine 129 (K129), which affects its homodimerization, binding to Crm1, and nuclear export [126]. Survivin can also be modified via K48- and K63-linked ubiquitination during mitosis, which mediates survivin localization to the centromere and mitotic progression [118].

## 7. Therapeutic Strategies to Target Survivin

Efforts have been made in recent years to develop therapeutic strategies to disable survivin functionally. However, suvivin is an unconventional drug target, due to its unique structure and lack of enzymatic activity. It is important to include cell-based assays that evaluate the phenotypic changes affected by inhibition of survivin. Because survivin is required for cell division and knockdown of survivin generally causes mitotic failure [32,58,59,67,68], it is plausible that the gross inhibition of survivin by small molecules would lead to similar effects. Currently, only a limited number of survivin inhibitors have been developed with success.

YM155, an imidazolium-based compound identified by a high-throughput screen, is one of the first small-molecule antagonists known for its ability to inhibit survivin expression. YM155 targets the survivin promoter region to inhibit gene transcription [127]. Preclinical research showed that YM155 can effectively decrease survivin expression levels in various cancer cell lines and inhibit tumor growth in xenografts mouse models, including prostate cancer, non-Hodgkin lymphoma, and lung cancer [127,128]. Several phase I and phase II studies showed that YM155 generally shows low toxicity but has limited antitumor efficacy when used as either a single agent or in combination with other therapeutic agents [129,130,131,132,133,134]. However, recent studies indicate that YM155 can elicit DNA damage in cells [135,136,137], which indicates that the compound may target other proteins and signaling events. Indeed, when tested in vitro, YM155 induces cell death without causing any delay in mitosis, despite that it significantly reduces survivin expression [135,136]. These findings suggest that the primary mechanism by which YM155 induces cytotoxicity is likely not through disabling survivin functions.

Several other molecules that suppress survivin expression levels have been described. For example, FL-118 has been identified by an HTS screen of a library of compounds as a molecule that can reduce expression levels of survivin, as well as those of other IAP family members [138]. FL-118-induced cell death accompanies a reduction in BrdU-incorporating cells, but does not with any effect on mitosis [138]. By using a similar approach to screen for drugs that can inhibit survivin promoter activities, a cytotoxic molecule, termed WM127, was also found to be capable of reducing survivin expression levels [139]. WM127 reduces cancer cell proliferation and causes an accumulation of cells in the G2/M stage of the cell cycle [139], although further analysis of its effect on mitotic progression remains to be carried out. In addition, EM-1421 (also known as terameprocol) has been described as a small molecule that targets Sp1-dependent promoters and reduces the expression levels of survivin and the mitosis regulator cdk1 [140,141,142]. Moreover, GDP366 is another small molecule that can reduce survivin expression at both mRNA and protein levels and increase aberrant cell division and polyploid cells [143]. However, because GDP366 can also inhibit the expression of stathmin 1 [143], which encodes a protein that mediates the dynamics of the microtubule network and mitotic progression [144,145,146], the mechanism of action by this molecule remains to be clarified. Thus, these efforts to target survivin expression have identified molecules that are not specific for survivin.

Strategies designed to reduce survivin protein stability have been reported. Heat shock protein 90 can bind to survivin and protect it from proteasomal degradation, and the disruption of this interaction can lead to apoptosis [147]. Sheperdin, a peptidomimetic derived from the survivin region that is sufficient to bind to Hsp90, showed the ability to bind the ATP pocket of HSP90 and disrupt the interaction with several of its client proteins, including survivin [148]. This causes the degradation of survivin, among other proteins, leading to apoptosis in tumor cells [148]. Of note, sheperdin treatment caused rapid cell death without triggering any apparent delay in mitosis [148].

In another example, using a virtual screen of compounds that mimic the DIABLO/SMAC-IAP interaction, a series of small-molecule IAP inhibitors have been developed [149,150]. These molecules can inhibit survivin and, to a lesser extent, XIAP, by downregulating their protein levels and showed efficacy to inhibit tumor growth [149,150]. Other small molecules that block the interaction between survivin and DIABLO/SMAC have been described and showed anti-cancer activities [151,152].

A high-throughput, affinity-based NMR screen has led to the identification of several survivin-binding molecules that bind to the dimer interface [153,154,155,156]. Several of the compounds identified in the screen displayed activities to inhibit the growth of tumor cells and appeared to cause cell cycle delay in the G0/G1 stage, rather than in the mitotic stage [154]. One of the molecules was shown to sensitize colon cancer cells to topoisomerase inhibitor irinotecan [156].

We employed a unique structure-based approach to identify survivin inhibitors that bind directly to the protein and modulate its functions [157,158]. The method, termed Cavity-Induced Allosteric Modulation (CIAM), was previously used to successfully develop inhibitors for other targets, such as TNFR1 [159]. With the CIAM method, we have identified a cavity close to the survivin dimeric interface. The compounds that fit into this cavity in silico were further tested for the ability to bind the survivin protein and affect mitotic arrest, as one would expect from a loss of survivin function. Several compounds identified by this approach, including S12, exhibit efficacy to inhibit the growth of human cancer cell lines both in vitro and in vivo [26,43,157,158]. This was the first set of small-molecule inhibitors that have been shown to directly bind to the intended target site in the survivin protein and cause phenotypic changes in cancer cell cells that are consistent with what is expected from loss of survivin function. Notably, the knockdown of YAP can increase the sensitivity of cancer cells to S12 [158], which suggests that simultaneously targeting survivin and YAP may achieve enhanced therapeutic effects. Finally, S12 has been modified to be potentially used for imaging survivin in tumors by single-photon emission computerized tomography [160].

More recently, a separate group performed a virtual screen of molecules that target the survivin dimeric interface and identified a series of molecules (e.g., LQZ-7F and LQZ-7I) that can induce the proteasomal degradation of survivin [161,162]. These molecules were also shown to cause apoptosis and inhibit tumor growth in xenograft tumor models [161,162]. It is not clear how disruption of the survivin homodimer by these small molecules leads to a reduction in protein stability. The effect of these molecules on cell cycle progression has not been well characterized [161,162].

Survivin-based immunotherapy has been developed. The Cytotoxic T lymphocyte (CTL) response to survivin can be detected in patients [163]. The survivin-derived, MHC-restricted T cell epitope has been identified and can be harnessed to trigger the CTL response against survivin-expressing cancer cells [164]. In addition, DNA vaccine-encoding survivin and CCL12 can trigger a strong immune response against lung cancer cells in an animal model [165]. Recent studies showed that long synthetic peptides derived from the survivin protein can generate both cytotoxic CD8+ and CD4+ T-cell responses, leading to tumor regression and the prevention of relapse in animal models [166]. In particular, the survivin peptide mimic SurVaxM showed efficacy in stimulating anti-tumor immune responses against brain tumors in animal models and early promise in clinical trials [167,168,169]. A whole protein survivin dendritic cell vaccine has also been developed and tested for the treatment of myeloma patients [170,171].

## 8. Conclusions and Perspective

Because of its high expression levels in most tumors and absence in most normal tissues, survivin has been considered a promising therapeutic target. Studies in the past 25 years have established that survivin plays a vital role in the regulation of cell division as a component of the CPC in the mitotic apparatus. However, despite its structural similarities to the other IAP family proteins, how survivin acts as an inhibitor of apoptosis remains elusive. The overexpression of survivin may lead to aberrant localization of the protein, which can contribute to aspects of tumorigenesis in a cell content-dependent manner. Studies on survivin protein ensembles in various subcellular pools, such as the mitochondria and the interphase or mitotic cytoplasm, may unravel a mechanism by which survivin links mitotic checkpoint regulation with apoptotic pathways. Progress has been made to develop survivin-targeted therapy, including small molecules that directly bind and disable survivin. With the advent of artificial intelligence-aided structural modeling and drug design, it is expected that more survivin-targeting entities will be available for testing. It should be mentioned that well-designed biophysical analysis and cell-based assays are critical for the identification of small molecules that elicit more specific anti-survivin activities. Finally, understanding the signaling pathways that determine cancer cell sensitivity to survivin-targeted therapy may help to develop more effective therapeutic strategies. Conceivably, the combination of survivin inhibition with chemotherapy or other targeted therapeutics may achieve the maximal clinical benefit.

## Figures and Tables

**Figure 1 cancers-16-01705-f001:**
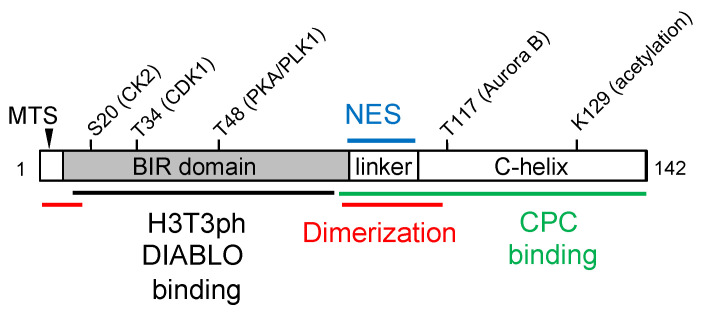
Schematic representation of survivin structure features involved in dimerization (red), chromosome passenger complex (CPC) binding (green), histone H3 threonine 3 phosphorylated peptide (H3T3ph) and DIABLO binding (black), nuclear export (NES, blue), and mitochondria targeting (MTS, arrowhead). The amino acid residues that have been reported to be modified by acetylation or phosphorylation, as well as the kinases involved, are also indicated.

**Figure 2 cancers-16-01705-f002:**
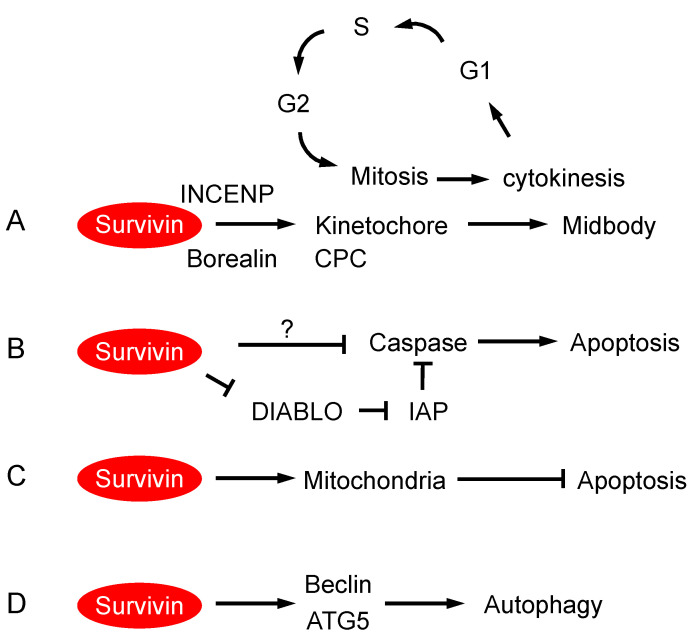
Functions of survivin. (**A**) Survivin is required for mitosis and cytokinesis. Survivin is associated with INCENP and borealin as components of the chromosome passenger complex (CPC) localized to the centromere during mitosis. Survivin remains associated with the spindle midbody from the anaphase of mitosis until the end of cytokinesis. (**B**) Survivin can protect cells from apoptosis. The binding of survivin to DIABLO may prevent the latter from inactivating other inhibitor of apoptosis (IAP) family proteins. Alternatively, survivin may directly inhibit caspase activity. (**C**) Survivin can be localized to the mitochondria and protect cells from mitochondria-mediated apoptosis. (**D**) Survivin can be associated with beclin or ATG5 and is postulated to be involved in aspects of autophagy.

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
