# Peer review of "Survivin as a Therapeutic Target for the Treatment of Human Cancer"

_cancers, 2024, doi:10.3390/cancers16091705_

Round 1
Reviewer 1 Report
Comments and Suggestions for Authors
This is a review article summarizing the biological roles of survivin and strategies to inhibit survivin as a means to control cancers. Overall the review is well balanced and covers all key aspects on survivin. I have two suggestions:
1. The two figures come in too late in the article. I feel they might be more helpful if introduced earlier.
2. The integration and interpretation of information is there but seems not less sufficient for a good review article. For example, which functions of survivin (anti-apoptosis, mitosis, mitochondria and autophagy) might be most important for a specific cancer? Or what happens to the different functions of survivin for a specific inhibition strategy? It's most likely the functions of survivin might be context dependent not only based on cancer tissue types but also other genetic and epigenetic alterations. It would be great if the authors can dissect even one example in certain depth instead of just listing previous observations.
Author Response
This is a review article summarizing the biological roles of survivin and strategies to inhibit survivin as a means to control cancers. Overall the review is well balanced and covers all key aspects on survivin. I have two suggestions:
- The two figures come in too late in the article. I feel they might be more helpful if introduced earlier.
We have revised the manuscript to show the figures earlier.
- The integration and interpretation of information is there but seems not less sufficient for a good review article. For example, which functions of survivin (anti-apoptosis, mitosis, mitochondria and autophagy) might be most important for a specific cancer? Or what happens to the different functions of survivin for a specific inhibition strategy? It's most likely the functions of survivin might be context dependent not only based on cancer tissue types but also other genetic and epigenetic alterations. It would be great if the authors can dissect even one example in certain depth instead of just listing previous observations.
We appreciate the suggestions from the reviewer and have revised the manuscript accordingly. We agree with the reviewer that certain functions of survivin may be dependent on cell context. We have revised the manuscript to include this point.
Reviewer 2 Report
Comments and Suggestions for Authors
The review of Qiang Wang and Mark I. Greene, is dedicated to investigations the role of survivin in promoting malignancy and the strategies for development of survivin-targeted therapy for cancer. They discussed that how survivin mediates various aspects of malignant transformation and drug resistance, as well as efforts that have been made to develop therapeutics targeting survivin for the treatment of cancer. The work should be published taking into account the following comments.
Major comments:
1. The review text is incorrectly structured. The sequence of review sections should be changed.
2. What is novelty of this review, it should be emphasized more clearly.
(for example, there is a similar review published earlier -
Yamamoto T, Tanigawa N. The role of survivin as a new target of diagnosis and treatment in human cancer. Med Electron Microsc. 2001 Dec;34(4):207-12. doi: 10.1007/s007950100017. PMID: 11956993).
Minor comments:
1. What is the reason for the choice of following processes….”cell division, apoptosis, mitochondrial function and autophagy”? Is there any information about survivin role in regulation the other processes?
2. It is not so clear. Information described in Chapter “Therapeutic strategies to target survivin” is actually only for malignant diseases?
Author Response
The review of Qiang Wang and Mark I. Greene, is dedicated to investigations the role of survivin in promoting malignancy and the strategies for development of survivin-targeted therapy for cancer. They discussed that how survivin mediates various aspects of malignant transformation and drug resistance, as well as efforts that have been made to develop therapeutics targeting survivin for the treatment of cancer. The work should be published taking into account the following comments.
Major comments:
- The review text is incorrectly structured. The sequence of review sections should be changed.
We have revised the structure of the text based on the template file provided by the journal.
- What is novelty of this review, it should be emphasized more clearly.
(for example, there is a similar review published earlier -
Yamamoto T, Tanigawa N. The role of survivin as a new target of diagnosis and treatment in human cancer. Med Electron Microsc. 2001 Dec;34(4):207-12. doi: 10.1007/s007950100017. PMID: 11956993).
We appreciate the comment from the reviewer and revised the manuscript accordingly. Our main goal is to provide a more up-to-date and critical review of the literature on the biology of survivin and the efforts to develop therapeutics. Almost all the studies on survivn target therapy were published after the 2001 paper by Yamamoto et al. In our manuscript, we provide a roadmap for development more effective and specific surviving-targeted molecules.
Minor comments:
- What is the reason for the choice of following processes….”cell division, apoptosis, mitochondrial function and autophagy”? Is there any information about survivin role in regulation the other processes?
We decided to follow this sequence because we believe there is more solid and convincing data that support the role of surviving in cell division. The role of survivin in the other biological events is more controversial due to lack of more direct evidence. We have emphasized this point in the revised manuscript.
- It is not so clear. Information described in Chapter “Therapeutic strategies to target survivin” is actually only for malignant diseases?
We have revised the manuscript to clarify that the scope of our review is within malignant diseases.